# Deconstructing S-Duality

**Kyle Aitken$^{1\star}$, Andreas Karch$^1$ and Brandon Robinson$^2$**

**1** Department of Physics, University of Washington, Seattle, Wa, 98195-1560, USA
**2** School of Physics & Astronomy and STAG Research Centre, University of Southampton, Highfield, Southampton SO17 1BJ, UK

$\star$ kaitken17@gmail.com

## Abstract

We use the technique of deconstruction to lift dualities from 2+1 to 3+1 dimensions. In this work we demonstrate the basic idea by deriving S-duality of maximally supersymmetric electromagnetism in 3+1 dimensions from mirror symmetry in 2+1. We also study the deconstruction of a non-supersymmetric duality in 3+1 dimensions using Abelian bosonization in 2+1.



## 1  Introduction

Recent years have seen tremendous progress in our understanding of dualities in field theories in 2+1 dimensions even in the absence of supersymmetry. For Abelian gauge groups a rich

web of dualities can be derived from one simple base pair [1,2], and some powerful dualities also exist in the non-Abelian case [3–5]. While some potential applications arise in the context of quantum Hall physics [6], the impact of these recent developments on our understanding of nature is still somewhat hampered by the fact that these low-dimensional dualities do not describe the 3+1 dimensional world we live in. Unfortunately, as far as 3+1 dimensional dualities in interesting gauge theories are concerned, our understanding has not evolved much beyond the realm of supersymmetry.

One may wonder whether some of the recent progress in 2+1 dimensions can be lifted to 3+1 dimensions. One promising avenue to pursue is the idea of deconstruction [7]. Deconstruction allows one to write a product gauge theory in $d$ dimensions which, in an intermediate energy regime, mimics a gauge theory in $d + 1$ dimensions. The Lagrangian of the $d$ dimensional gauge theory is essentially the $d + 1$ dimensional theory put on a circle of radius $R$ with the compact direction discretized on a one dimensional lattice of lattice spacing $a$ with $N$ sites so that $2\pi R = Na$. At energies $a^{-1} \gg E \gg R^{-1}$ the theory behaves as a $d + 1$ dimensional continuum field theory. In the UV the lattice spacing becomes visible and the theory reverts to the $d$ dimensional product gauge group. At energies below $R^{-1}$ the theory of course also becomes $d$ dimensional due to the circle compactification.

In this work we will deconstruct maximally supersymmetric electromagnetism that is a simple $\mathcal{N} = 4$ supersymmetric $U(1)$ gauge theory. The 2+1 dimensional field theory accomplishing this is a $U(1)^N$ quiver gauge theory with bifundamental matter and $\mathcal{N} = 4$ supersymmetry.[1] Such supersymmetric 2+1 dimensional quivers are well known to have a dual description [8]: Mirror symmetry maps the quiver to supersymmetric QED with $N$ flavors. Given that the quiver gauge theory realizes 3+1 dimensional maximally supersymmetric electromagnetism, mirror symmetry implies that QED with $N$ flavors also has to have a 3+1 dimensional window, at least when the parameters are dialed appropriately. In this work we will demonstrate that this indeed the case. The mirror theory in 2+1 dimensions realizes the electromagnetic dual, or S-dual, of the 3+1 dimensional theory we started with. A summary of the various theories and their relations is shown in figure 1.

Many of the techniques used in deconstructing 3+1 dimensional S-duality easily import into the non-supersymmetric case. From a base duality, we derive new all scale quiver dualities consisting of Wilson-Fisher scalars (WF) on one side and QED with $N$ flavors on the other; both theories containing Chern-Simons terms. The quiver now deconstructs a non-supersymmetric $U(1)$ gauge theory in 3+1 dimensions, which implies QED with $N$ flavors must also have a 3+1 dimensional description. Up to order one factors hidden by strong coupling, this again appears to be the case, and it further motivates the conjecture that 2+1 dimensional bosonization realizes non-supersymmetric electromagnetic duality of the original 3+1 dimensional theory.

The organization of this note is as follows. In section 2 we will review the essentials of deconstruction and, in particular, exhibit the 2+1 dimensional deconstruction of maximally supersymmetric 3+1 dimensional electromagnetism. In section 3 we recall the basics of mirror symmetry. We will see that in order to construct the mirror of deconstructed $\mathcal{N} = 4$ electromagnetism we need its all scale version [9]. In section 4 we present our main result: we demonstrate that the mirror theory indeed reproduces the S-dual realization of 3+1 dimensional electromagnetism in the continuum limit. In section 5, we consider a non-supersymmetric version of deconstructing 3+1 dimensional Abelian dualities through the all scale version of bosonization. We conclude in section 6.

---

[1]Note that $\mathcal{N} = 4$ supersymmetry in 2+1 dimensions is half the supersymmetry of the 3+1 dimensional theory we are aiming for, 8 instead of 16 supercharges. This is a well known phenomena in deconstruction: together with the broken translation invariance along the circle direction due to the discretization we lose half of the supersymmetry for every compact dimension. The full supersymmetry of the 3+1 dimensional theory only emerges in the continuum limit, that is at energies below $a^{-1}$.

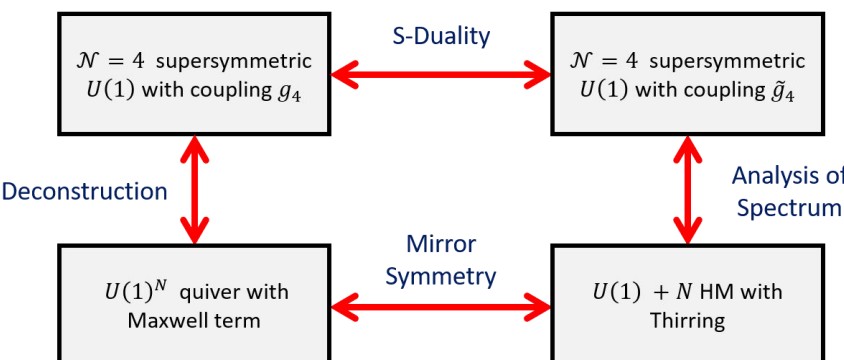

Figure 1: Summary of the relationship between the various supersymmetric theories we consider.

## 2   Deconstruction

The action for a pure 3+1 dimensional gauge theory with one of the spatial dimensions compacitified on a circle of circumference $2\pi R$ and discretized into $N$ lattice sites of lattice spacing $a$ reads [7]

$$S = \int d^3x \left( -\frac{1}{4G^2} \sum_{i=1}^{N} F_i^2 + \frac{1}{2G^2 a^2} \sum_{i=1}^{N} (D_\mu U_{i,i+1})^\dagger (D^\mu U_{i,i+1}) + \dots \right). \tag{1}$$

Here $i$ labels the $N$ lattice sites. There is a 2+1 dimensional gauge theory with field strength $F_i = d b_i$ living on each site with the same gauge group as the 3+1 dimensional parent. In this work we will be mostly concerned with the Abelian case, but deconstruction works equally well for any gauge group. From the 2+1 dimensional point of view we are describing a $U(1)^N$ gauge theory. $G$ is the 2+1 dimensional gauge coupling on each site with its standard dimension of Length$^{-1/2}$. The diagonal $U(1)$ decouples completely since no matter is charged under it, but the corresponding photon is part of our dynamical fields. This is different from how quiver gauge theories are typically treated in discussions of mirror symmetry, where the diagonal $U(1)$ gauge boson is removed from the spectrum completely and one only keeps the non-trivial $U(1)^N/U(1)$ part of the gauge theory.

The group valued matrices $U$ live on the links, so they are labeled by $(i, i+1)$, the two sites they connect. From the 3+1 dimensional point of view the kinetic term for $U$ simply is a discretization of $-F_{\mu 3} F^{\mu 3}/(4G^2)$ and so the two terms combine into the standard gauge kinetic term. From the 2+1 dimensional point of view the $U$ fields can be thought of as "non-linear sigma model fields" in the bifundamental representation. Under a gauge transformation $\mathfrak{g}_i$ the $U$ fields transform as

$$U_{i,i+1} \rightarrow \mathfrak{g}_i^{-1} U_{i,i+1} \mathfrak{g}_{i+1}. \tag{2}$$

The covariant derivatives of $U_{i,i+1}$ are given by

$$D_\mu U_{i,i+1} \equiv \partial_\mu U_{i,i+1} - i b_\mu^i U_{i,i+1} + i U_{i,i+1} b_\mu^{i+1}. \tag{3}$$

For our Abelian case we can think of the $U$'s as pure phase variables,

$$U_{i,i+1} = e^{i a \phi_{i,i+1}}. \tag{4}$$

$\phi$ can be interpreted as $b_4$, the component of the gauge field along the circle direction. The non-linear sigma models described by the $U$ fields are not UV complete theories in $d = 2+1$ dimensions all by themselves. The non-linear sigma model has a strong coupling scale,

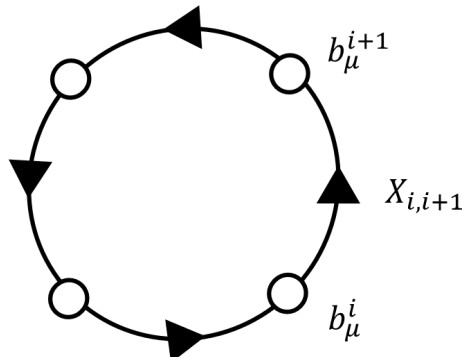

Figure 2: Quiver diagram for the 2+1 dimensional deconstruction of maximally supersymmetric electromagnetism. Nodes (white circles) correspond to $U(1)$ gauge groups, links (black arrows) to bifundamental hypermultiplets.

$f_\pi^{-1/2} = aG$. But the non-linear sigma models can be UV completed entirely within the 2+1 dimensional theory. The original work of [7] envisioned two scenarios to do so. One option is to introduce an extra quiver node in the middle of every link. Furthermore, the new link fields will just be bifundamental fermions. As long as all the extra nodes are dialed to confine at an energy scale $\Lambda \gg a^{-1}$, the relevant degrees of freedom at scale $a^{-1}$ are simply the pions of that confining gauge group, which are well described as non-linear sigma model fields. The alternate option, studied in [10], is to replace the non-linear sigma model fields with linear sigma model fields, that is with un-constrained bifundamental scalar fields $X_{i,i+1}$. As long as we force the scalar fields to have a vacuum expectation value

$$\langle X_{i,i+1} \rangle = X_0, \tag{5}$$

the low energy degrees of freedom are once more given by the phase variables, $X_{i,i+1} = X_0 e^{ia\phi_{i,i+1}}$. In general this second option requires one to carefully choose quartic potentials to ensure that we get the desired $X_0$. However, in supersymmetric gauge theories, especially those with extended supersymmetry, this second option is in fact very easy to implement. Such theories often have moduli spaces of vacua: the scalar potential is exactly flat and we simply can dial the vacuum expectation value $X_0$ as a free parameter.

Using this second purely 2+1 dimensional UV completion, we can present the deconstruction of a pure $\mathcal{N} = 4$ supersymmetric $U(1)$ gauge theory in 3+1 dimensions. The relevant theory is a 2+1 dimensional $\mathcal{N} = 4$ quiver gauge theory with $N$ nodes connected according to the quiver diagram in figure 2. Each node corresponds to a $U(1)$ vector multiplet: a dynamical photon together with two adjoint Dirac fermions and three adjoint real scalars. Since we are considering only $U(1)$ gauge groups, adjoint here simply means neutral. Each link corresponds to a bifundamental hypermultiplet, that is 2 complex scalars and 2 Dirac fermions with charge $(+1, -1)$ under the two $U(1)$ factors they connect.

Besides $N$, the basic parameters are $G$, the coupling of each $U(1)$ factor, and $X_0$, the vacuum expectation value for scalar fields in the bifundamental hypermultiplets. This theory indeed has a moduli space of vacua, so $X_0$ can freely be dialed and different values of $X_0$ correspond to different super-selection sectors. More precisely, $X_0$ is a parameter along the Higgs branch of the theory, where the gauge group gets broken down to the diagonal subgroup,

$$U(1)^N \to U(1). \tag{6}$$

The moduli space in principle also contains a Coulomb branch where $X_0$ vanishes but instead the scalars in the vectormultiplets living on the nodes get a vacuum expectation value. The Coulomb branch will make an appearance in the dual theory, since mirror symmetry maps

the Higgs branch of one theory to the Coulomb branch of the other. But to use the standard deconstruction procedure we need to be on the Higgs branch.

This quiver theory deconstructs a 3+1 dimensional pure $\mathcal{N} = 4$ gauge theory with $U(1)$ gauge group on a circle of radius $R$ with parameters of the 3+1 theory given by

$$g_4^2 = \frac{G}{X_0}, \quad 2\pi R = \frac{N}{GX_0}, \quad a = 2\pi\frac{R}{N} = \frac{1}{GX_0}. \tag{7}$$

The identification of $a$ comes from looking at the kinetic term for the scalar. The mass term the gauge bosons pick up from the Higgs mechanism reads $X_0^2(b_\mu^{i+1} - b_\mu^i)^2$. Comparing this to the desired action (1) tells us that we need to identify $X_0^2 = (aG)^{-2}$. The circumference $2\pi R$ is $Na$, the number of lattice sites times the lattice spacing. To identify the 4d gauge coupling we can simply look at the massless mode of the gauge field. From the 3+1 dimensional point of view this is the constant zero mode along the circle, so its coupling is given by performing the integral over the circle in the action and hence $g_{ZM}^{-2} = (2\pi R)g_4^{-2}$. From the 2+1 dimensional point of view the zero mode is the unbroken $U(1)$ that remains after breaking $U(1)^N \to U(1)$, so its coupling is given by $g_{ZM}^{-2} = NG^{-2}$. Equating the two expressions for the zero mode gives us the identification of $g_4$ in terms of $G$ and $X_0$. This construction follows exactly the one outlined in [11] where maximally supersymmetric gauge theories in 4+1 and 5+1 dimensions were deconstructed in terms of 3+1 dimensional gauge theories.

As described in the introduction, the theory behaves 3+1 dimensional in the energy window $a^{-1} \gg E \gg R^{-1}$. To take a genuine continuum limit we want to send $a \to 0$ while keeping $g_4$ and $R$ fixed. So we need:

$$G, X_0, N \to \infty, \quad \frac{G}{X_0} \text{ fixed}, \quad \frac{N}{GX_0} \text{ fixed}. \tag{8}$$

One important aspect of this limit is that we can not take $G$ to infinity from the get go. Since the ratio of $G$ and $X_0$ is fixed, $a = (GX_0)^{-1}$ is really the only basic length scale in the problem. If we work at strictly infinite coupling, this scale vanishes identically and so we never see a finite radius circle. We only want to take $a$ to zero as we take $N$ to infinity. Infinite $G$ corresponds to the IR limit of the theory. In order to understand what happens to our theory as we take the limit we need to be able to control excitations away from the strict IR limit.

Since the 3+1 dimensional theory lives on a circle, the only massless gauge boson is the zero mode, in full agreement with the fact that at low energies we only have a single 2+1 dimensional $U(1)$ gauge theory. Note that this low energy theory is actually $\mathcal{N} = 8$ supersymmetric, that is it has 16 supercharges, since it is just the circle reduction of maximally supersymmetric electromagnetism. The fact that we are really studying a 3+1 dimensional theory on a circle is encoded in the spectrum of Kaluza-Klein (KK) modes. Since $X_0$ is the vacuum expectation value for bifundamental matter we see that the mass matrix for the massive W-bosons takes the standard balls-and-springs form

$$M_W^2 = G^2 X_0^2 \begin{pmatrix} 2 & -1 & 0 & \cdots & 0 & -1 \\ -1 & 2 & -1 & 0 & & 0 \\ 0 & -1 & 2 & -1 & & \vdots \\ \vdots & 0 & -1 & \ddots & & 0 \\ 0 & & & & 2 & -1 \\ -1 & 0 & \cdots & 0 & -1 & 2 \end{pmatrix}. \tag{9}$$

with eigenvalues

$$m_n^2 = 4G^2 X_0^2 \sin^2 \frac{\pi n}{N}, \tag{10}$$

where $n$ is an integer with $-N/2 < n \leq N/2$. These are in fact the expected momentum modes on a discretized circle. Our classical calculation of the W-masses unfortunately is only valid in the weak coupling limit, whereas our continuum limit involves a strong coupling limit. In a supersymmetric theory one might hope that the W-masses could be protected by supersymmetry. As pointed out in [11] this is almost true here: while the W-bosons are *not* BPS states in the full quiver gauge theory, they do become BPS states from the point of view of the low energy $\mathcal{N} = 8$ gauge theory. What this means is that while we can not guarantee that the full spectrum of W-boson states remains to be given by (10) at strong coupling, we can trust the formula for the low lying states with $n \ll N$ where $m_n^2 \approx n^2/R^2$ exactly as we expect in the continuum limit. We have in fact successfully deconstructed maximally supersymmetric electromagnetism in 3+1 dimensions.

Let us close this section with noting that there is an alternate way of thinking of the parameter $X_0$. We can force the bifundamental scalar fields to have a non-vanishing expectation value by adding an Fayet-Iliopoulos (FI) terms to the theory.[2] A FI is a term linear in the auxiliary $D$ field in the vector multiplet, $\mathcal{L}_D = \xi D/\pi$. After integrating out $D$ it corresponds to a new scalar term in the potential together with the fermionic partners that are needed for supersymmetry. In terms of the moduli space, the FI term imposes a "D-term" constraint that ensures that the modified potential vanishes. A FI term $\xi_i$ on any of the individual nodes would give a D-term demanding that the expectation values of the 2 complex scalar fields $X_{i,i+1}$ and $\tilde{X}_{i,i+1}$ on each site obey

$$|X_{i,i+1}|^2 - |\tilde{X}_{i,i+1}|^2 - |X_{i-1,i}|^2 + |\tilde{X}_{i-1,i}|^2 - \frac{\xi_i}{\pi} = 0. \tag{11}$$

This is not what we want. It would force a non-vanishing difference between the expectation values of the incoming and outgoing link at each node. We do want to work with $\xi_i = 0$ for all $i$ so that all scalars can have the same expectation value $X_0$. Fortunately there is a different $U(1)$ symmetry we can use in the problem. Rotating all matter fields by an overall phase,

$$X_{i,i+1} \rightarrow e^{i\alpha/N} X_{i,i+1}, \qquad \tilde{X}_{i,i+1} \rightarrow e^{-i\alpha/N} \tilde{X}_{i,i+1}, \tag{12}$$

gives a global symmetry rather than a gauge symmetry. We can introduce a background FI term for this global symmetry as a parameter in our Lagrangian, giving us a D-term

$$\frac{1}{N} \sum_i (|X_{i,i+1}|^2 - |\tilde{X}_{i,i+1}|^2) - \frac{\xi}{\pi} = 0. \tag{13}$$

For $\xi = \pi X_0^2$ this gives us the desired $X_{i,i+1} = X_0$ as the simplest solution. Thinking of $X_0$ as being imposed on us by a FI term will help us with the parameter mapping to the mirror.

# 3 All Scale Mirror Symetry

## 3.1 Mirror Symmetry

Mirror symmetry as introduced in [8] is an IR duality, that is it is only valid at infinite $G$. Mirror symmetry relates two different supersymmetric gauge theories that flow to the same conformal field theory in the infrared. In terms of deformations due to giving scalar fields vacuum expectation values, it maps the Coulomb branch of one theory to the Higgs branch

---

[2]In our conventions we follow [9], where all normalizations are clearly spelled out. The Maxwell term has a prefactor of $(4G^2)^{-1}$, Chern-Simons like terms have an integer coefficient times $(4\pi)^{-1}$, whereas FI terms have a coefficient of $\pi^{-1}$.

of the other and vice versa. While mirror symmetry is well established also for non-Abelian gauge groups [8, 12, 13], the general dual pair is easiest to construct in the Abelian case. For Abelian mirror symmetry all possible dual pairs can be derived from a single base pair [9]. This is in complete parallel with the derivation of the non-supersymmetric duality web [1, 2] from a single seed duality. In fact, it was the supersymmetric insights that served as a template for the non-supersymmetric web. In the $\mathcal{N} = 4$ supersymmetric case the base pair demands the following equivalence:

**Base pair (BP):** $U(1) + 1$ charged hyper = free hyper.

Both sides of the base pair have a global $U(1)$ symmetry and the corresponding conserved current can be coupled to a background source $B_\mu$. Here and in what follows we will denote dynamical gauge fields with lower case letters and background fields with upper case letters.[3] We use $a_\mu$ for the dynamical gauge field on the left hand side of (BP) and denote its field strength by $f_{\mu\nu}$.

On the right hand side of (BP) we see a free hypermultiplet. The global symmetry is simply particle or baryon number, counting the number of scalars and fermions. The corresponding background field couples via the standard minimal coupling in the kinetic term. On the left hand side baryon number is gauged, so it is not a global symmetry. Instead we have a topological global symmetry, monopole number, whose conserved current is $j_\mu = \epsilon^{\mu\nu\rho} f_{\nu\rho}$. $j^\mu$ is identically conserved due to the Bianchi identity. The background field couples to the theory via a BF term

$$S_{BF}[a, B] = \frac{1}{2\pi} \int d^3x \, \epsilon^{\mu\nu\rho} a_\mu \partial_\nu B_\rho. \tag{14}$$

Once again, this is a story that has recently resurfaced in the context of non-supersymmetric dualities but had been well appreciated in the supersymmetric context long ago.

Last but not least let us recall how we can use (BP) to derive mirror duals for arbitrary Abelian gauge theories. Starting with $N$ copies of the right hand side we can realize any Abelian gauge theory by promoting $r$ of the $N$ global baryon number symmetries to gauge symmetries. Which linear combination we gauge is up to us, so we can chose our $N$ matter fields to have charges $R_i^a$ under the $r$ gauge groups. This construction has been nicely laid out in [14]. We end up with

**Theory A:** $\mathcal{N} = 4$ $U(1)^r$ gauge theory with $N$ hypermultiplets of charge $R_i^a$.

Here $i$ runs from 1 to $N$ as before, whereas $a$ runs from 1 to $r$. The dual theory starts with $N$ copies of the left hand side, that is $N$ copies of SUSY QED with 1 flavor coupled to background fields via BF couplings. Promoting $r$ of the background gauge fields to dynamical gauge fields gives masses to both the $r$ background gauge fields and the $r$ dynamical gauge fields they couple to via the BF couplings. We are left with the $N - r$ gauge groups (labeled by $p$ running from 1 to $N - r$), that is with

---

[3]Readers familiar with the non-supersymmetric duality web may wonder whether our gauge and background fields are genuine $U(1)$ connections or rather spin$_c$ connections. For the supersymmetric duality all our background fields are in fact $U(1)$ connections and not spin connections. Symmetries under which a whole multiplet is charged manifestly violate the spin/charge relation that is required to hold for spin$_c$ connections. Both fermions and bosons in the multiplet carry identical charges. The only exception to this rule are R-symmetries, under which the super-charges themselves are charged and so bosons and fermions within a given multiplet transform differently. The $\mathcal{N} = 4$ supersymmetric gauge theories we consider have an $SU(2)_L \times SU(2)_R$ R-symmetry and treating background fields for these symmetries as spin$_c$ connections would allow us to study the theories on manifolds which aren't spin but only spin$_c$. For our purposes here, turning on background fields for R-symmetries is not needed and so all our dynamical and background fields are $U(1)$ connections.

**Theory B:** $\mathcal{N} = 4$ $U(1)^{N-r}$ gauge theory with $N$ hypermultiplets of charge $S_i^p$ with $\sum_i R_i^a S_i^p = 0$ for all $a$ and $p$.

The quiver is a special case of this general duality. Usually it is considered to be the $r = N - 1$ case with charges assigned according to the quiver of figure 2 with the overall $U(1)$ modded out (so that the $N$ nodes only give rise to $N - 1$ gauge groups). The overall $U(1)$ completely decouples from the theory. It can be added back in on both sides in the end to obtain our deconstructing $U(1)^N$ theory of figure 2. The dual is simply a single $U(1)$ gauge group with $N$ flavors, that is $\mathcal{N} = 4$ supersymmetric QED. This duality between the Abelian quiver theory with $N$ nodes and QED with $N$ flavors was in fact the very first example of mirror symmetry uncovered in [8].

This particular instance of mirror symmetry can also be very easily understood in terms of Hanany-Witten style brane setups [15]. Introducing branes as displayed in table 1, we can suspend snippets of D3 branes between NS5 branes in the $x_6$ direction. As long as this direction is compactified on a circle, we can realize the quiver gauge theory of figure 2 via a single D3 brane wrapping the entire $x_6$ circle with $N$ NS5 branes along the circle as displayed in the left panel of figure 3.

|      | 0 | 1 | 2 | 3 | 4 | 5 | 6 | 7 | 8 | 9 |
|------|---|---|---|---|---|---|---|---|---|---|
| NS5  | x | x | x | x | x | x | o | o | o | o |
| D3   | x | x | x | o | o | o | x | o | o | o |
| D5   | x | x | x | o | o | o | o | x | x | x |

Table 1: Brane realization of 2+1 dimensional quiver gauge theories and their mirrors.

Each D3 segment connecting neighboring NS5 branes corresponds to a $U(1)$ gauge group factor. The bifundamental hypermultiplets arise from strings connecting D3 brane segments across the NS5 branes. The coupling constants $G_i$ of the individual $U(1)$ factors is encoded in the length $L_i$ of the D3 segments, since the 2+1 dimensional gauge theories arise from compactifying the 3+1 dimensional gauge theory living on the D3 branes on the segment, so $G_i^{-2} = L_i g_4^{-2}$. The configuration in figure 3 corresponds to the theory which we need for deconstruction with the NS5 branes equally spaced along the circle so that all gauge couplings are equal, $G_i = G$ for all $i$. This is not the point at which mirror symmetry is valid. The conformal field theory which has the two mirror symmetric descriptions is only realized when all $N$ NS5 branes sit on top of each other. The equally spaced configuration can be viewed as a deformation of the CFT by the irrelevant Maxwell terms corresponding to the NS5 brane separations. Once again we see that for the application to deconstruction we would like to understand the theory away from the strict IR limit.

To find the mirror dual from the brane setup we simply perform S-duality in type IIB string theory. This takes the D3 brane back into itself, but exchanges NS5 branes with D5 branes. To stay within the configurations made out of the objects displayed in table 1 we need to combine this transformation with a relabeling of the $x_{3,4,5}$ and $x_{7,8,9}$ directions. This relabelling captures the fact that mirror symmetry exchanges the two $SU(2)$ R-symmetry factors, which correspond to rotations in 789 and 345 respectively. In the field theory the two R-symmetries act on Higgs and Coulomb branch respectively and so exchanging them also means we are exchanging the two branches. The dual of the quiver gauge theory is displayed in the right panel of figure 3. This time there are no NS5 branes present, so we have a single $U(1)$ gauge group. The $N$ D5 branes give rise to $N$ charged hypermultiplets. So indeed we see that the mirror is QED with $N$ flavors.[4] We saw that in the quiver gauge theory, the CFT point corresponds to

---

[4]As noted before, the $U(1)$ gauge theory living on the D3 brane is in fact an $\mathcal{N} = 8$ gauge theory, supersym-

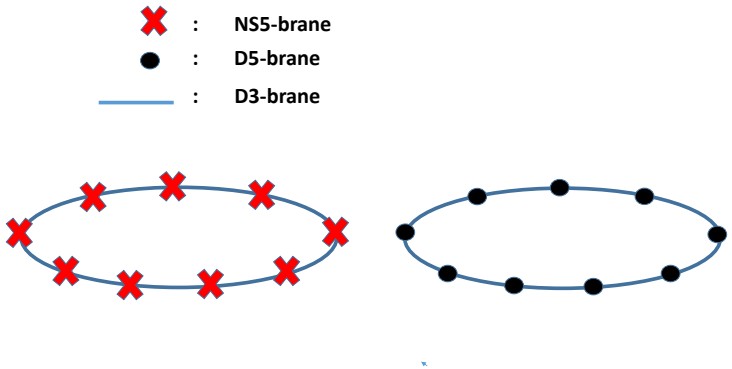

Figure 3: Brane construction of the quiver gauge theory and its mirror dual.

coincident NS5 branes and the configuration relevant for deconstruction involves turning on the irrelevant deformations corresponding to the NS5 separations. In the electric quiver theory these corresponded to the gauge couplings, the concrete identification of the corresponding "magnetic couplings" on the mirror side has long been known to be non-trivial. Fortunately in the Abelian case a full resolution is known, as we will explain in the next subsection.

Let us close this discussion with noting that the brane pictures we just introduced immediately generalize to the non-Abelian case. If instead of a single D3 brane we use $K$ D3 branes, the quiver gauge theory will be $U(K)^N$ with bifundamental matter. The overall $U(1)$ again decouples, but this time all $N$ $SU(K)$ factors are interacting. This quiver theory nicely deconstructs $\mathcal{N} = 4$ SYM with $U(K)$ gauge group. The mirror dual theory is $U(K)$ with $N$ flavors. The main reason we restrict ourselves to the Abelian case in this work is the identification of the magnetic couplings which we are about to review.

## 3.2 All Scale Duality

Mirror symmetry as it stands is only valid in the deep IR, that is when the $U(1)$ factors are infinitely coupled. We have argued before that in order to properly understand the dual of deconstructed maximally supersymmetric electromagnetism, we seem to require being able to keep track of both theories even at finite coupling. Fortunately this is easy to do, at least in the Abelian case, using the same techniques that we used to derive new mirrors from a single base pair. We can start with (BP), gauge the global $U(1)$ baryon number on the right hand side, but this time add a $F^2/(4G^2)$ Maxwell term to it, with $F$ the field strength of $B_\mu$. Since $B_\mu$ started out as a background gauge field, we should feel free to add such contact terms before we promote it to a dynamical gauge field, $B_\mu \to b_\mu$, as long as we do so consistently on both sides of the duality. This gives us the theory on the left hand side of (BP) but with an extra Maxwell term. Whatever this same procedure does on the original left hand side of (BP) has got to be the new right hand side, that is the all-scale dual to $U(1) + 1$ charged hyper + Maxwell term. On the original left hand side of (BP) the dual monopole $U(1)$ couples via a BF term, so we have, schematically

$$S_{\text{lhs}} = \text{Hyper}[a] + S_{BF}[a, b] + \tfrac{1}{4G^2} \int F^2.$$

metry is only broken to $\mathcal{N} = 4$ by the charged matter hypermultiplets. That is, there is an additional adjoint (=neutral) hypermultiplet present. In the $U(1)$ case, this adjoint hypermultiplet is completely decoupled. A free hypermultiplet is equivalent to a free vector multiplet (after dualizing the photon into a dual scalar) and it is this extra hypermultiplet that, in the brane picture, is dual to the decoupled overall $U(1)$ on the electric side. The non-trivial $U(1)^N/U(1)$ quiver gauge theory is dual to $\mathcal{N} = 4$ QED with $N$ flavors.

Hyper[$a$] denotes the action of the hypermultiplet minimally coupled to $a_\mu$ and $S_{BF}$ is the BF coupling of (14). Integrating out $a_\mu$ and $b_\mu$ gives a (supersymmetric version) of the Thirring model: a hyper with a quartic fermion interaction. This is irrelevant in 2+1 dimensions, but so is a Maxwell term. The Thirring interaction is the dual to the Maxwell term. After exchanging left and right hand side, this gives an all-scale version of (BP):

**all scale Base pair (aBP):**

$$U(1) + 1 \text{ charged hyper} + \text{Maxwell} = \text{free hyper} + \text{Thirring}.$$

While (aBP) is useful as a mnemonic to understand what all scale mirror symmetry does, we will find it more useful in what follows to keep both the massive gauge fields around on the Thirring side of the duality in order to correctly capture the physics of this irrelevant deformation.

Applying this philosophy to the $U(1)^N$ quiver gauge theory we want for deconstruction, we need to start with $N$ free hypermultiplets and gauge $N$ linear combinations of the global baryon number symmetries. Denoting the original baryon number symmetries by $B_i$, we form the bifundamental linear combinations

$$B_i = b_{i+1} - b_i, \tag{15}$$

and also promote the $b_i$ to dynamical fields. Here the index $i$ is defined modulo $N$, that is $b_{N+1} \equiv b_1$. At this step we can introduce a Maxwell term with coupling $G$ for each of the $U(1)$ factors $b_i$. Repeating the same steps as in the case of a single $U(1)$ above, we can find the all scale dual for the quiver, including the irrelevant coupling $G$. The dual has $N$ QED gauge bosons $a_i$ from the $N$ decoupled factors of QED with 1 flavor we started out with. These couple to the same $N$ dynamical gauge fields $B_i = b_{i+1} - b_i$ we newly gauged. Somewhat schematically, the mirror action (M) reads

$$S_{\text{M}} = \sum_i \text{Hyper}_i[a_i] + \sum_i S_{BF}[a_i, b_{i+1} - b_i] + \sum_i \frac{1}{4G^2} \int F_i^2. \tag{16}$$

Once again, we could integrate out the massive gauge fields $a_i$ and $b_i$ to reduce to QED with $N$ flavors (and a decoupled center of mass factor). The coupling constant $G$ would imprint itself as a current-current Thirring like interaction. But for the purpose at hand it will prove more convenient to directly work with (M).

## 4 S-duality from mirror symmetry

In the previous section we identified the mirror (M) of the quiver gauge theory responsible for deconstructing maximally supersymmetric electromagnetism in 3+1 dimensions. We found that at the conformal point the non-trivial part of the dual is just QED with $N$ flavors. Turning on the irrelevant Maxwell coupling $G$ induces a particular pattern of current-current interactions which is best captured by the introduction of an extra $2N-2$ massive gauge fields. There is one more step we need to take in order to implement the theory described by the matching in (7): we need to turn on the vacuum expectation value $X_0$ for the scalars in the quiver theory. Recall that in the original quiver this expectation value took us out onto the Higgs branch, where $U(1)^N$ is broken to the diagonal $U(1)$. In the dual theory (M) this same expectation value takes us out onto the Coulomb branch. If we integrate out the massive vectors and simply look at QED with $N$ flavors going on the Coulomb branch amounts to giving masses to all flavors. Writing $\mathcal{N} = 4$ in $\mathcal{N} = 2$ notation, the adjoint chiral multiplet $X$ that is part of the

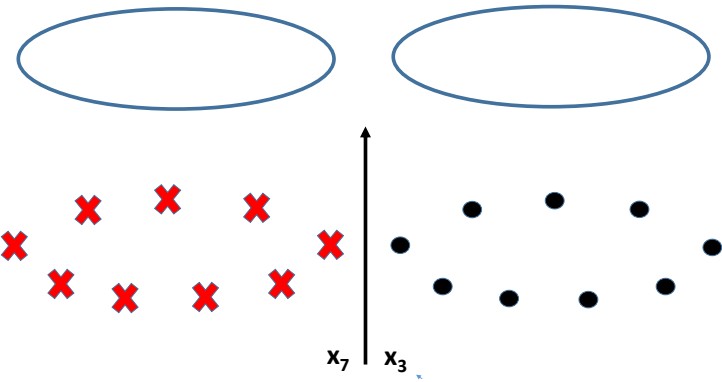

Figure 4: Higgs branch deformation of the quiver gauge theory and the dual Coulomb branch deformation of its mirror.

$\mathcal{N} = 4$ vector multiplet couples to the chiral multiplets $\tilde{Q}_i$ and $Q_i$ in the hypermultiplet via a superpotential coupling

$$W \sim X \sum_i \tilde{Q}_i Q_i. \tag{17}$$

That is giving a vacuum expectation value $\langle X \rangle = X_0$ to the vector multiplet scalar is equivalent to adding mass $X_0$ to all $N$ matter fields.

The effects of moving out onto this branch, both in the original and in the mirror theory, can once again be best understood from a brane picture as displayed in figure 4.

Turning on $X_0$ to move onto the Higgs branch in the original quiver corresponds to moving the D3 brane off from the NS5 branes into an orthogonal direction (say $x_7$). This brane picture makes it manifest that the low energy physics is just the 3+1 dimensional $\mathcal{N} = 4$ $U(1)$ gauge theory living on the D3 brane. The continuum limit is exactly achieved by taking the limit where the D3 brane gets infinitely removed. The KK modes living on the D3 brane appear as W-bosons from the UV field theory. In the S-dual picture we see the same low energy physics: a single D3 brane far removed (this time in the $x_3$ direction) from a stack of D5 branes. Since the two pictures are related by type IIB S-duality, which acts as electro-magnetic duality on the D3, the $U(1)$ on the D3 brane this time is the S-dual of the theory we originally deconstructed.

The D-brane realization of the gauge theory also allows us to make one more important point. In addition to the W-bosons we expect from the deconstructed circle, our theory also has BPS vortices. In the string theory construction they correspond to D1 strings connecting the D3-brane to one of the NS5 branes. In the mirror theory they map to F1 fundamental strings connecting the D3 to one of the D5s. The mass of these states is proportional to the length of the string, so it grows linearly with $X_0$. In the continuum limit, where $X_0 \to \infty$, these states become infinitely heavy and decouple.

To confirm that the lessons the brane picture teach us about the mirror are indeed realized in the field theory we need to confirm that the low energy physics of theory (M) along its Coulomb branch indeed realizes a tower of KK modes. Instead of integrating out the massive vector bosons in (M) to study the spectrum of the massive large $N$ Thirring-like model that is left behind, we find it more convenient to integrate out the $N$ massive matter fields instead. After all, the limit we are taking is the one in which the flavor fields get infinitely heavy. Integrating out the flavor fields in the action (16) leaves behind, at low energies, $2N$ vector bosons $a_i$ and $b_i$ with masses induced from the topological BF terms. The effects of the massive flavors is to modify the kinetic terms for the gauge fields they couple to, that is the $a_i$ fields. This is encoded in an effective metric on the Coulomb branch, which fortunately is tightly

constrained due to the high supersymmetry in the system and we can directly borrow the results from [16, 17] to read off the answer.

To understand exactly how $X_0$ is related to the hypermultiplet mass it is easiest to use the FI terms to force the expectation value. $\xi$ is an FI term for the D-term of the background gauge field coupling to the baryon number global symmetry. On the magnetic side this symmetry is mapped to the monopole number of the overall $U(1)$ field

$$\tilde{a} = \frac{1}{N} \sum_i a_i. \tag{18}$$

$\tilde{a}$ couples to the background field via a BF term whose supersymmetric completion couples the scalar superpartner $\tilde{x}$ of $\tilde{a}$ to the auxiliary field $D$ of the background multiplet. Together with the FI term this means that the terms in the action containing $D$ simply read $D(\tilde{x} - \xi)/\pi$; so $D$ acts as a Lagrange multiplier that forces $\tilde{x}$ to get expectation value equal to $\xi$. Or in other words, the hypermultiplets pick up mass $m = \xi$. This is the standard parameter mapping in mirror symmetry: FI terms map to mass terms. So our task is to determine the correction to the gauge couplings of the gauge fields under which the hypermultiplets are charged due to integrating out a massive hypermultiplet of mass $m = \xi = \pi X_0^2$. The hypermultiplets are neutral under the $b_i$ gauge fields, so the latter are unaffected and their gauge coupling remains $G$. Each $a_i$ has exactly one charged hypermultiplet. The quantum corrections to QED with 1-flavor are indeed well understood [16,17]. This theory has no Higgs branch (quartic potentials prevent the hypermultiplets from getting expectation values) but has a one (quaternionic) dimensional Coulomb branch. Corrections are usually phrased in terms of the metric on this branch which encodes the kinetic terms in the low energy effective action along the branch. Classically the Coulomb branch is simply $\mathbb{R}^3 \times S^1$, parametrized by the 3 scalars in the vector multiplet together with the dual photon. Quantum mechanically the Coulomb branch picks up a 1-loop exact correction turning it into Taub-NUT [16]. This metric has no singularities, which would signal extra massless particles, in accordance with string theory expectations [17]. But the correction can be interpreted as including a shift in the effective gauge coupling which has been shown in [18] to read (in the normalization of [9] we have been using all along):

$$g_{a_i}^{-2}(X) = g_{cl}^{-2} + \frac{1}{4\pi|\vec{X} - \vec{m}|}, \tag{19}$$

where $\vec{X}$ is the triplet of the scalars contained in the hypermultiplet and $\vec{m}$ is the triplet of the allowed supersymmetric mass terms. Note that the classical value of the gauge coupling, $g_{cl}$, is not $G$. In the derivation of all-scale mirror symmetry adding a Maxwell term on the electric side with coupling $G$ induced a Thirring term or equivalently a Maxwell term for $b_i$, but not for $a_i$. The latter remained infinitely strongly coupled throughout, $g_{cl}^{-2} = 0$. The $U(1)$ gauge couplings of the $a_i$ fields is entirely due to integrating out the hypermultiplets of mass $m = \pi X_0^2$:

$$g_{a_i} = 2\pi X_0. \tag{20}$$

With this the mirror action (M') on the Coulomb branch reads

$$S_{\text{M'}} = \sum_i \frac{1}{4(2\pi X_0)^2} \int f_i^2 + \sum_i S_{BF}[a_i, b_{i+1} - b_i] + \sum_i \frac{1}{4G^2} \int F_i^2, \tag{21}$$

where $f_i$ is the field strength of $a_i$. To calculate the spectrum of massive fluctuations we can rescale $f_i \to f_i' = \frac{2\pi X_0}{G} f_i$ so that all $2N$ gauge fields have the same kinetic term. The newly rescaled fields of course would not be properly quantized, but for the purposes of determining the spectrum of small fluctuations this is irrelevant. Maxwell's equations for the $2N$ gauge

fields pick up an effective mass matrix from the $BF$ terms, which can be written in block diagonal form

$$M_M = G X_0 \begin{pmatrix} 0 & C \\ C^T & 0 \end{pmatrix}. \tag{22}$$

0 denotes an $N$ by $N$ block vanishing entries and $C$ is the $N$ by $N$ matrix given by

$$C = \begin{pmatrix} +1 & -1 & 0 & \cdots & 0 & 0 \\ 0 & +1 & -1 & 0 & & 0 \\ 0 & 0 & +1 & -1 & & \vdots \\ \vdots & & 0 & \ddots & & 0 \\ 0 & & & & +1 & -1 \\ -1 & 0 & \cdots & 0 & 0 & +1 \end{pmatrix}. \tag{23}$$

Note, in particular, that comparing to the W-boson mass matrix of (9) we have

$$C^T C = \frac{M_W^2}{G^2 X_0^2}. \tag{24}$$

So the eigenvalues of the magnetic mass matrix $M_M$ indeed are just the square root of the W-boson mass squareds from (10), that is

$$m_n^2 = 4 G^2 X_0^2 \sin^2 \frac{\pi n}{N}, \tag{25}$$

where once again $n$ in an integer with $-N/2 \le n \le N/2$. Note that in $M_M$ each eigenvalue shows up twice. That is due to the fact that we are really working out the masses of a single polarization of a gauge field, whereas $M_W^2$ was already fully accounting for the 2 polarizations of the massive gauge boson after "eating" the corresponding scalars. So the spectrum of massive KK modes completely agrees, the mirror Thirring theory indeed grew an extra dimension!

Looking at the massless modes, we see two massless vectors on the mirror side. One of those has all $a_i = \tilde{a}$ with $b_i = 0$ and the other one has all $b_i = \tilde{b}$ with $a_i = 0$. On the electric side we had a massless vector and a massless hyper (the "Higgs boson") combining into an $\mathcal{N} = 8$ vector multiplet. As we alluded to before, a decoupled massless vector can be rewritten as a massless hyper and, indeed, the two sides are completely equivalent. As we pointed out above, the fact that we needed to dualize a single free hyper into a vector is also manifest in the brane picture.

Last but not least let us look at the coupling constants in the magnetic theory. The coupling constant $G$ plays the role of a magnetic coupling in the mirror theory, as expected. The electric coupling is what governs the Maxwell term of the zero mode $\tilde{a}$ from (18), which was induced from the massive hypermultiplets. Since each $a_i$ had coupling $X_0$, we have that the corresponding coupling of this zero mode is given by

$$\tilde{g}_{ZM}^{-2} = N(2\pi X_0)^{-2}, \tag{26}$$

from which we can obtain the gauge coupling of the deconstructed 3+1 dimensional theory as before:

$$\tilde{g}_4^2 = (2\pi R)\tilde{g}_{ZM}^2 = (Na)\frac{(2\pi)^2 X_0^2}{N} = (2\pi)^2 \frac{X_0}{G} = \frac{(2\pi)^2}{g_4^2}. \tag{27}$$

As expected[5] [19], the coupling constant of the dual theory is $(2\pi)^2$ times the inverse of the coupling of the electric theory! We successfully deconstructed S-duality.

---

[5]Note that the $g^2$ as defined in [19] is twice the coupling squared we have been using here.

# 5   Deconstruction with Abelian bosonization

Let us take above analysis and apply it to a more conjectural setting: non-supersymmetric Abelian dualities. Taking stock of the key ingredients needed in the supersymmetric case, we are motivated to consider the Abelian duality

$$\textbf{BP}': \quad \text{WF Scalar} = \text{Fermion} + U(1)_{-\frac{1}{2}}$$

as a starting point. Quivers valid in the IR limit can be constructed in a manner similar to those found in [20]. The philosophy behind deriving the all-scale Base Pairs remains intact. That is, all of the quantities appearing in the process of arriving at the aBP relation used in deconstruction are simple supersymmetric generalizations of terms that can be easily identified in the non-supersymmetric context: The $\mathcal{N} = 4$ $U(1)$ gauge theories nodes with hypermultiplet links of the quiver become Chern-Simons theories coupled by bifundamental Wilson-Fisher (WF) scalars, and the dual picture becomes $N$ flavors of Dirac fermions with flux attachment in the form of a $U(1)_{-\frac{N}{2}}$ Chern-Simons theory. That is, the single node non-supersymmetric version of aBP reads

$$\textbf{aBP}': \quad \text{WF Scalar} + (U(1) + \text{Maxwell}) = U(1)_{-\frac{1}{2}} + \text{Fermion} + (U(1) + \text{Maxwell}).$$

The main concern is what becomes of the Chern-Simons terms in the quiver theory in the 3+1 dimensional limit. Naïvely, one would guess that they become some form of linearly varying $\theta$-angle [21]. To understand what happens, let us consider the case where all of the scalars acquire the same vacuum expectation value $\langle X_{i,i+1} \rangle = X_0$ such that we again arrive a linear sigma model for the phase variable, $U_{i,i+1}$, i.e. $X_{i,i+1} = X_0 U_{i,i+1}$. The discretized scalar theory becomes

$$S_{\text{scalar}} = k^{ij} S_{CS}[b_i; b_j] + \int d^3 x \left( -\frac{1}{4G^2} \sum_{i=1}^{N} F_i^2 + \frac{1}{2G^2 a^2} \sum_{i=1}^{N} \left( D_\mu U_{i,i+1} \right)^\dagger \left( D^\mu U_{i,i+1} \right) + \dots \right), \tag{28}$$

where we use

$$S_{CS}[A_i, A_j] \equiv \frac{1}{4\pi} \int d^3 x \, \epsilon_{\mu\nu\rho} A_i^\mu \partial^\nu A_j^\rho \tag{29}$$

to denote both Chern-Simons and BF terms and the coefficients take on a form reminiscent of the balls-and-springs mass matrix we saw earlier

$$k^{ij} = \begin{pmatrix} 2 & -1 & 0 & \cdots & 0 & -1 \\ -1 & 2 & -1 & 0 & & 0 \\ 0 & -1 & 2 & -1 & & \vdots \\ \vdots & 0 & -1 & \ddots & & 0 \\ 0 & & & & 2 & -1 \\ -1 & 0 & \cdots & 0 & -1 & 2 \end{pmatrix}. \tag{30}$$

The form of (28) is very similar to the supersymmetric case with the exception of the Chern-Simons terms, which we will now argue have no effect on the continuum deconstructed theory. The discretized Chern-Simons term take the form of a combination of forward and backward hopping terms on the lattice:

$$k^{ij} S_{CS}[b_i; b_j] = \frac{a}{4\pi} \sum_{i=1}^{N} \left( \frac{b_i - b_{i-1}}{a} \right) d b_i - \frac{a}{4\pi} \sum_{i=1}^{N} \left( \frac{b_{i+1} - b_i}{a} \right) d b_i, \tag{31}$$

where periodicity $b_i \equiv b_{i+N}$ has been imposed. What is immediately obvious is that in the continuum limit the two hopping terms become equivalent and cancel one another. Thus despite being important for the functioning of the quiver duality, the Chern-Simons and BF terms do not affect the duality in the limit to a continuum theory in 3+1 dimensions.[6] Indeed, if the bifundamental scalars all pick up the same vacuum expectation value, then the dynamical gauge bosons acquire the same mass matrix as in (9), and taking limit as in (8), we find simply that the scalar side of the duality has deconstructed pure 3+1 dimensional Maxwell theory compactified on an $S^1$

$$S = -\frac{1}{4g_4^2} \int_{\mathbb{R}^3 \times S^1} d^4x \, F_b^2, \tag{32}$$

with $F_b$ the 3+1 dimensional field strength. Note that our non-supersymmetric theory no longer has a moduli space, so $X_0$ is not a parameter we can dial. We expect that adding a large negative mass squared to all Wilson-Fisher scalars will force them to pick up a large expectation value, but due to the the strong coupling limit involved we have no control over the order one factors.

The question now is what becomes of the fermionic side of the quiver dual picture in the continuum limit, which also comes with additional Chern-Simons terms,

$$S_{\text{fermion}} = -\sum_i S_f[\psi_i, a_i] - \frac{\delta^{ij}}{2} S_{CS}[a_i; a_j] - \delta^{ij} S_{CS}[a_i; b_j - b_{j+1}] - \sum_i \frac{1}{4G^2} \int F_i^2. \tag{33}$$

On the scalar side, the continuum limit corresponds to a large negative mass squared for the matter fields and, through the identification of mass deformations in the single node context $m_\psi \leftrightarrow -m_X^2$, we expect that continuum limit in the fermionic theory corresponds to opening a large gap in the fermion spectrum. Integrating out heavy fermions in QED$_3$ not only cancels out their respective $\eta$-invariants but also gives to one loop [18, 22] the coupling of the 3+1 dimensional theory

$$g_{a_i}^{-2} = g_{cl}^{-2} + \frac{1}{12\pi m_\psi}. \tag{34}$$

We note again that clasically $g_{cl}^{-2} \to 0$ as Abelian bosonization is, like BF, a relation in the deep IR, and so $g_{a_i} = \sqrt{12\pi m_\psi}$. The gapped theory is then described by

$$S'_{\text{fermion}} = -\sum_i \frac{1}{4g_{a_i}^2} \int f_i^2 - \delta^{ij} S_{CS}[a_i; b_j - b_{j+1}] - \sum_i \frac{1}{4G^2} \int F_i^2. \tag{35}$$

Canonicalizing the kinetic terms, diagonalizing the mass matrix for the gauge fields, and computing the mass spectrum of Kaluza-Klein modes gives

$$m_n^2 = \frac{G^2 g_{a_i}^2}{\pi^2} \sin^2 \frac{\pi n}{N}. \tag{36}$$

Analogous to the supersymmetric case, we can postulate $m_\psi \sim X_0^2$.[7] Then, with (34), $g_{a_i}^2 \sim X_0^2$ and hence (36) is the same as (25), up to an $\mathcal{O}(1)$ factor. Similar arguments follow for the

---

[6]One might also worry that the additional Chern-Simons terms in the 2+1 dimensional theory will give competing mass contributions to $b_i$. Although the Chern-Simons induced masses scale as $G^2 \sim GX_0$ in the continuum limit, the Chern-Simons mass matrix also comes with two factors of $k^{ij}$. Diagonalizing each factor of $k^{ij}$ gives $\sin^2(\pi n/N) \sim n^2/N^2$ for $n \ll N$. Thus, the Chern-Simons masses are $1/N^2$ suppressed relative to those due to Higgsing. This is consistent with the 3+1 dimensional point of view that effects due to the Chern-Simons terms are suppressed in the continuum limit.

[7]At least for negative scalar mass squared this is the more basic version of the parameter map in bosonization, where the fermion mass is directly identified with the scalar expectation value.

non-supersymmetric generalization of (27), i.e. $\tilde{g}_4^2 \sim (2\pi)^2 g_4^{-2}$. So while the basic construction seems to work just as in the supersymmetric case, it is difficult to confirm this duality quantitatively. It's not even clear that the spectrum of KK modes survives the strong coupling limit unless one can find a large $N$ argument that suppresses corrections.

# 6  Discussion

Let us take stock of what has been accomplished. We used a well established yet still somewhat conjectural duality between strongly coupled field theories in 2+1 dimensions in order to rederive a somewhat trivial statement about a free theory in 3+1 dimensions. What is this good for? For one, the procedure we followed is a nice additional check of mirror symmetry, in particular its somewhat less well studied all-scale version. But, of course, more interesting is that this demonstration can serve as the existence proof that potentially new dualities in 3+1 dimensions can indeed be derived from established dualities in 2+1.

A fairly straightforward generalization of the work we have done here is to study the non-Abelian case in order to establish the less trivial S-duality of $\mathcal{N} = 4$ SYM. The brane construction is identical to the one we used here and so we expect exactly the same pattern of masses in this case. The new complication is that this time we do not know of a Lagrangian description of the magnetic coupling. Unlike the Abelian $U(1)$ monopole number symmetry, which is visible at all scales and can be coupled to via a BF term, the corresponding non-Abelian symmetries only arise as accidental symmetries in the IR. As long as we take a formal definition of the magnetic coupling as an irrelevant deformation of the IR fixed point, the brane picture demonstrates that the mirror theory will again grow an extra dimensions if we chose the electric theory to deconstruct $SU(K)$ SYM in 3+1 dimensions.

One other obvious generalization would be to include a $\theta$ term in the gauge theory; one would hope that it should be possible to find the full $SL(2,\mathbb{Z})$ invariance of the 3+1 dimensional theory and not just S-duality.

More interesting will be to return to the recent progress in our understanding of 3d bosonization dualities. While the original set of dualities were established for theories with single gauge groups [3] the generalization to Abelian [20] and even non-Abelian product gauge groups [23], including those based on quivers, appears to be straightforward. In either case, the 2+1 dimensional dualities typically include Chern-Simons terms which aren't present in the straightforward application of deconstruction and their role needs to be clarified. For the Abelian case we were able to demonstrate that the basic construction carries through, even though the lack of control over the strong coupling regime makes it difficult to make quantitative comparisons in this case. The non-Abelian case presents extra complications. Beyond the problems with the magnetic couplings we already encountered in the supersymmetric case, the flavor bounds inherent in the dualities of [3] make it difficult to derive dualities for some of the most interesting non-Abelian quivers we might want to study. Nevertheless, the fact that in the simplest, most supersymmetric example deconstruction allowed us to derive a well established 3+1 dimensional duality from 2+1 dimensional mirror symmetry makes us hopeful that progress along these lines can be made.

## Acknowledgments

We'd like to thank Kristan Jensen, Shamit Kachru, and David Tong for useful correspondence as well as Andrew Baumgartner and Michael Spillane for useful conversations. The work of KA and AK was supported, in part, by the U.S. Department of Energy under Grant No. DE-

SC0011637. The work of BR was funded, in part, by STFC consolidated grant ST/L000296/1.

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
