# Peer review of "Deconstructing S-Duality"

_SciPost Physics, doi:SciPost Phys. 4, 032 (2018)_

## Round 1 · Referee Report · Anonymous · 2018-3-30

Strengths
1 - Includes a nice review of deconstructed extra dimensions.
2 - Provides a number of arguments in support of conjectured (though mostly well established) dualities in 2+1 and 3+1 dimensions, and relates those dualities via deconstruction.
3 - Explains how to generalize certain IR dualities to all scales.
4 - Backs up claims from several perspectives.
5 - Points out the additional challenges faced by the non-Abelian generalization of the analysis by the authors, and suggest a path forward in that direction.
Weaknesses
1 - This is a nice paper all around. A minor annoyance is that the paper introduces numerous acronyms.
Report
This is quite a nice paper detailing relations between a class of 2+1 and 3+1 dimensional dualities of Abelian gauge theories. The main idea is that the 3+1 dimensional gauge theories can be deconstructed into 2+1 dimensional theories with product gauge group. The 2+1 dimensional quiver theories are equivalent to the corresponding 3+1 dimensional theories for some range of energies.
The authors carefully demonstrate the relationship between mirror symmetry in the 2+1 dimensional theory and S-duality in the corresponding 3+1 dimensional theory. They match the two sets of dualities both from a field-theoretic analysis and from the corresponding Type IIB brane construction of the deconstructed theory. They then attempt to derive an analogous relation between dualities in a nonsupersymmetric version of the same theories, but this discussion is less convincing than in the supersymmetric setting. They also briefly discuss challenges in constructing a non-Abelian generalization of their analysis, but also suggest a path forward.
This paper is well thought through, backs up its claims with evidence from several perspectives, and provides new evidence for certain conjectured dualities. It is well written, except that it uses too many acronyms that complicate the readability of the text. There's BP, aBP, HM, WF, ZM...
Also, I personally find it helpful to refer to the number of supercharges rather than just to ${\cal N}$, at least the first time the theory is described. This is a minor point, but this addition would be useful for those of us who don't want to think too hard.
Finally, the authors suggest that it would be interesting to include the $\theta$ term in the gauge theory and understand the full SL(2,Z) duality. I agree. For this purpose, I wonder whether the lift of the brane configuration to M-theory would be helpful. In a similar context, Witten identified relative positions of the NS5-branes in the extra M-theory dimension with the theta parameters of the gauge groups.
In summary, I am strongly supportive of publication of this interesting paper, pending only optional suggested improvements.
Requested changes
1 - If at all possible, I advocate reducing the use of acronyms, especially new ones, as much as is feasible.
2 - I recommend listing the number of supersymmetries in addition to reference to ${\cal N}$=4 and so on.
3 - I would reference the paper by Hill, Pokorski and Wang (https://inspirehep.net/record/541000) when first mentioning deconstruction. They considered the linear sigma-model version of the story.

---

## Round 1 · Referee Report · Anonymous · 2018-4-1

Strengths
1- The main premise of the paper (combining 3D mirror symmetry and deconstruction) is a novel and interesting idea.
2- The exposition of the material is very pedagogical throughout.
Weaknesses
1- Only abelian supersymmetric theories are considered.
2- The final section on the extension to non-supersymmetric theories is rather conjectural.
Report
In this paper the authors combine dimensional deconstruction and three-dimensional mirror symmetry to obtain a field-theoretic derivation of four-dimensional S-duality. This is carried out for abelian, supersymmetric theories, although a brief discussion towards extending their approach to a non-supersymmetric setting using abelian bosonisation is also included.
At a technical level, the IR duality between two 3D theories (an abelian circular quiver and supersymmetric QED) is extended beyond infinite coupling to an “all-scale mirror symmetry”; this is important for the implementation of dimensional deconstruction, which requires taking an intricate limit of parameters that involves (but is not exclusive to) infinite coupling. The all-scale mirror dual of the 3D quiver is then considered on the Coulomb branch (deconstruction requires going to the Higgs branch but mirror symmetry exchanges the two), its 4D interpretation argued for and the candidate 4D abelian S-dual theories successfully compared.
The steps in the derivation of the above all-scale mirror symmetry for supersymmetric theories (known in the literature for some time) bear strong resemblance to those used in the more recent derivation of the non-supersymmetric duality webs for 3D theories. Such a connection could be the starting point for lifting 3D dualities to 4D using deconstruction, and this goal forms the main motivation behind this paper. Having said that, its implementation is technically difficult (as deconstruction involves strong coupling) and the statements towards the end of the article are of a rather conjectural nature.
Despite this shortcoming, and perhaps the fact that only abelian examples are tackled even in the supersymmetric case, this is an interesting paper which in my opinion deserves publication in SciPost. The exposition of both the background and new material is very pedagogical and excellent throughout.
Requested changes
None.

---

## Round 2 · Author Response

We thank both referees for their feedback and are glad they find our paper interesting and nice to read. Following the suggestions of referee 2 we made the changes outlined below.

---

## Round 2 · List of Changes

1 - In order to minimize the use of acronyms we removed all use of "HM" and made sure that the remaining acronyms are only used within equations (where we find them unavoidable) or when referring to equations, e.g. (BP) and (aBP) are names of equations and we do find it easier to read these letter labels rather than always referring to them by equation number.

2 - We are certainly sympathetic to the notion that when going between dimensions, the amount of supersymmetry is often better characterized by giving the total number of supercharges rather than using the "N =" counting of spinors. We had already paid tribute to this with our footnote 1, which translates the two most common players in our paper, N=4 SUSY in d=3+1 and d=2+1 into 16 and 8 supercharges respectively. To make sure all supersymmetries we use are at least once translated into this language, we added a few words after the first time we mention N=8 SUSY in 2+1 dimensions (which is also a 16 supercharge theory). While mostly using the "N=" notation throughout our paper, we now have every single theory we use defined in the language of total supercharges as well.

3 - We added a reference to the Hill, Pokorski and Wang paper when first discussing using linear sigma models for the purpose of deconstruction.

You are currently on this page

Resubmission 1802.01592v2 on 10 April 2018

---

## Editorial Decision

published